# Evolving Strategies and Materials for Scaffold Development in Regenerative Dentistry

Michal Gašparovič [1,*] , Petra Jungová [1,*] , Juraj Tomášik [1] , Bela Mriňáková [2] , Dušan Hirjak [3], Silvia Timková [4] , Ľuboš Danišovič [5] , Marián Janek [6,7], Ľuboš Bača [6], Peter Peciar [8] and Andrej Thurzo [1,*]

1 Department of Orthodontics Regenerative and Forensic Dentistry, Faculty of Medicine, Comenius University in Bratislava, 81102 Bratislava, Slovakia; tomasik7@uniba.sk

2 1st Department of Oncology, Faculty of Medicine, St. Elisabeth Cancer Institute, Comenius University in Bratislava, 81250 Bratislava, Slovakia; bela.mrinakova@fmed.uniba.sk

3 Department of Stomatology and Maxillofacial Surgery, Faculty of Medicine, Comenius University in Bratislava, 81250 Bratislava, Slovakia; dusan.hirjak@fmed.uniba.sk

4 Department of Stomatology and Maxilofacial Surgery, Faculty of Medicine, Pavol Jozef Safarik University and Louis Pasteur University Hospital, 04190 Kosice, Slovakia; silvia.timkova@upjs.sk

5 Institute of Medical Biology, Genetics and Clinical Genetics, Faculty of Medicine, Comenius University in Bratislava, 81108 Bratislava, Slovakia; lubos.danisovic@fmed.uniba.sk

6 Department of Inorganic Materials, Faculty of Chemical and Food Technology, Slovak University of Technology, 81237 Bratislava, Slovakia; marian.janek@stuba.sk (M.J.); lubos.baca@stuba.sk (Ľ.B.)

7 Department of Physical and Theoretical Chemistry, Faculty of Natural Sciences, Comenius University in Bratislava, 84215 Bratislava, Slovakia

8 Institute of Process Engineering, Faculty of Mechanical Engineering, Slovak University of Technology, 81231 Bratislava, Slovakia; peter.peciar@stuba.sk

* Correspondence: gasparovic58@uniba.sk (M.G.); jungova2@uniba.sk (P.J.); thurzo3@uniba.sk (A.T.)

**Abstract:** Regenerative dentistry has experienced remarkable advancement in recent years. The interdisciplinary discoveries in stem cell applications and scaffold design and fabrication, including novel techniques and biomaterials, have demonstrated immense potential in the field of tissue engineering and regenerative therapy. Scaffolds play a pivotal role in regenerative dentistry by facilitating tissue regeneration and restoring damaged or missing dental structures. These biocompatible and biomimetic structures serve as a temporary framework for cells to adhere, proliferate, and differentiate into functional tissues. This review provides a concise overview of the evolution of scaffold strategies in regenerative dentistry, along with a novel analysis (Bard v2.0 based on the Gemini neural network architecture) of the most commonly employed materials used for scaffold fabrication during the last 10 years. Additionally, it delves into bioprinting, stem cell colonization techniques and procedures, and outlines the prospects of regenerating a whole tooth in the future. Moreover, it discusses the optimal conditions for maximizing mesenchymal stem cell utilization and optimizing scaffold design and personalization through precise 3D bioprinting. This review highlights the recent advancements in scaffold development, particularly with the advent of 3D bioprinting technologies, and is based on a comprehensive literature search of the most influential recent publications in this field.

**Keywords:** regenerative medicine; scaffolds; mesenchymal stem cells; 3D printing; bioprinting; tissue engineering; whole tooth regeneration; large language model research; AI; LLM; Bard; ChatGPT

## 1. Introduction

### 1.1. Historical Background and Significance of Scaffold Development

Regenerative dentistry, with its high potential to address various dental defects, is a special part of regenerative medicine. The history of regenerative medicine and tissue engineering dates back to ancient times. Then, people held hopes and beliefs that living tissues or even whole organisms can be replaced or fabricated. More than 30 years ago, Langer and Vacanti introduced a new interdisciplinary field of tissue engineering [1]. The

idea from Greek mythology that "independent life can be created without sexual reproduction" may thus be considered the first recorded mention of creating living organisms from living or non-living specimens [2]. Nevertheless, it is claimed that the person to coin the term "regenerative medicine" was Leland Kaiser in 1992. In a hospital journal, he predicted a new field that would become a crucial part of algorithms for the management of chronic diseases and/or organ failures. Even before that, in 1985, Y.C. Fung introduced the idea of tissue engineering as an extended idea of biocompatibility [3,4]. Dentistry has been a part of the whole regenerative concept in medicine since the very beginning of its formation. In its essence, the scope of regenerative dentistry, albeit quite varied and diverse, is much smaller compared to the vast scope of regenerative medicine. That is partly the reason why such advances have been achieved. Another reason why novel therapies, including regenerative techniques, are applied in the craniofacial (and dental) area are their accessibility and smaller and less load-bearing nature of tissue defects, which all lead to the greater clinical application of the hereinbelow mentioned techniques [3,5].

Nowadays, regenerative dentistry is considered as a new specialization within dentistry. The approaches in this field have made a significant step forward over recent years, and we can expect that its further development will accelerate in the future and will be utilized in other medical fields. To ensure biocompatibility, adaptation, and proper regeneration of tissues, certain criteria need to be met, including functional complexity, neuromuscular coordination, and certain aesthetic characteristics [6–8]. For now, there have not been detailed instructions formulated that would vouch for successful living tissue replacement with no adverse effects. This narrative review provides the reader with up-to-date information and novel trends in the field of regenerative dentistry and tissue replacement therapy [9–12], including the most current utilization, applications, improvements, and methods over the last five years (2019–August 2023). To replace living tissue in the human body using biocompatible 3D-printed grafts (usually in the form of scaffolds), we need to understand numerous biological processes and interactions between native tissues, cells, and scaffolds, promote their migration towards the injured area, ensure adequate vascularization and biodegradation, etc. [8,13,14]. Nowadays, the market offers various specific materials with multiple advantageous properties, novel techniques and methods, including bioprinting and 3D printing options, and different types of stem cells. Furthermore, several growth factors, medications, signaling molecules, and other elements to stimulate regenerative processes have become widely available. There remains a need to further study materials and their combination, selection of stem cells, and additives, as well as all the processes throughout the construction and integration phase of scaffolds.

### 1.2. Timeline of Significant Advancements in Scaffold Development in Dentistry

There were various challenges in optimizing scaffold design, material selection, and compatibility with the oral environment, which were gradually overcome. The following timeline, summarized in Table 1, shows the significant advances in scaffold development over the last three decades. The ability to create custom-designed scaffolds with tailored properties promises immense advances in regenerative dentistry and in the restoration of damaged dental tissue.

Early 1990s

Initially, scaffolds were primarily composed of natural materials, such as collagen, silk, and alginate. These materials were biocompatible and biodegradable, but they offered limited control over pore size and architecture [1,5].

Late 1990s–Early 2000s

The introduction of synthetic polymers, such as polylactic acid (PLA) and polyglycolic acid (PGA) provided greater control over scaffold properties. These materials could be engineered with specific pore sizes and architectures to facilitate cell migration and tissue formation [15–17].

**Table 1.** The advancements in scaffold development over the past three decades.

| Decade | Scaffold Type | Key Features |
|---|---|---|
| Early 1990s | Natural Materials (collagen, alginate, silk, hyaluronic acid, chitosan) | Biocompatible, biodegradable |
| Late 1990s–Early 2000s | Synthetic Polymers (PLA, PGA) | Controlled pore size and architecture |
| Mid 2000s | Hybrid Scaffolds (various combinations of natural and synthetic polymers) | Biocompatibility, tunable properties |
| Late 2000s–Early 2010s | 3D Printed Scaffolds (HAp, TCP, PLA, PGLA, PCL, collagen) | Precision, complex architectures |
| Mid-2010s–Present (last decade) | Advanced materials (composites, hydrogels, bioactive materials) | Growth factors, signaling molecules, early explorations of using 4D materials [1] in scaffold development, bioactive agents, and other [2] |

[1] 4D materials can change their shape, properties, or function over time in response to stimuli, promoting dynamic tissue growth and healing. [2] Including 3D printing patient-specific scaffolds and development of decellularized scaffolds.

Mid-2000s

Hybrid scaffolds, which combined natural and synthetic materials, emerged as a promising approach. These scaffolds offered the biocompatibility and biodegradability of natural materials with the tunable properties of synthetic polymers [18,19].

Late 2000s–Early 2010s

The advent of 3D revolutionized the scaffold development, enabling precise control over scaffold geometry and material composition [20,21]. Bioprinting allowed for the creation of scaffolds with complex architectures that closely mimic native tissues, providing an optimal environment for tissue regeneration [21].

Mid-2010s–Present

Researchers are exploring the use of advanced materials, such as composites, hydrogels [22–25], and bioactive materials [26], to further enhance the performance of scaffolds. These materials may incorporate growth factors, signaling molecules, or other bioactive agents to promote cell adhesion, proliferation, and differentiation [27–29].

*1.3. Aim and Scope of This Review*

This narrative review focuses on the progress of scaffold approaches in regenerative dentistry. It is intended for researchers in this field to facilitate understanding of the context beyond their specific focus on material and method. As regenerative dentistry is just emerging, this review aims to provide a brief overview of the developmental steps that have led to the current state of the art, which could help researchers anticipate further trends in scaffold development. Another aim of this review was to perform literature research and analyze the most influential publications on this topic to examine the most commonly researched materials for scaffolds used in regenerative dentistry by novel utilization of large language models (LLMs) like Bard and ChatGPT.

The presented literature search aims to identify the most influential papers in the studied topic. It is based on the Field-Weighted Citation Impact (FWCI) indicator, and is limited to the last five years to capture recent advances in scaffold development. Therefore, it provides an overview of novel scaffold approaches in regenerative dentistry and excludes other aspects of regenerative medicine or dental tissue development as much as possible.

## 2. Materials and Methods

### 2.1. Evaluation of the Most Impactful Papers Based on Scaffold Manufacturing

The most impactful publications from 2019 to 2023, as identified through our literature search, explore the current progress in scaffold approaches in regenerative dentistry. The literature search was based on the Field-Weighted Citation Impact (FWCI) indicator, which is calculated for publications indexed in the SCOPUS database (Elsevier, Amsterdam, The Netherlands). This indicator evaluates the citation rate of documents throughout a three-year window, in comparison to similar documents, taking the year of publication, document type, and research field into account. The part of the review with the literature analysis was limited to the last 5 years and covers the most recent advances in scaffold development. This review focuses on scaffold approaches in regenerative dentistry and excludes other aspects of regenerative medicine or dental tissue engineering as much as possible.

The PubMed database was searched to identify relevant articles from 2019 to 2023. The search query was as follows:

*((scaffold[Title/Abstract]) OR (scaffolds[Title/Abstract])) AND ((regenerative dentistry [Title/Abstract]) OR (regenerative dental medicine[Title/Abstract]))*

The search was conducted on 3 August 2023. The titles and abstracts of searched articles were screened, as well as the key words. This search identified 43 relevant articles (Figure 1), all of which were further studied. The searched articles were divided into five subsection groups (Figures 1 and 2) according to the subsection topic that was most extensively covered in the article.

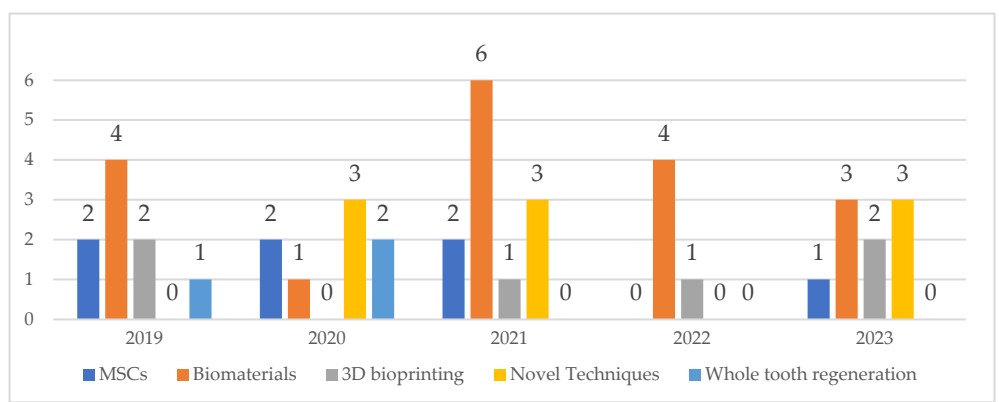

**Figure 1.** Number and type of articles selected for the review based on the search query.

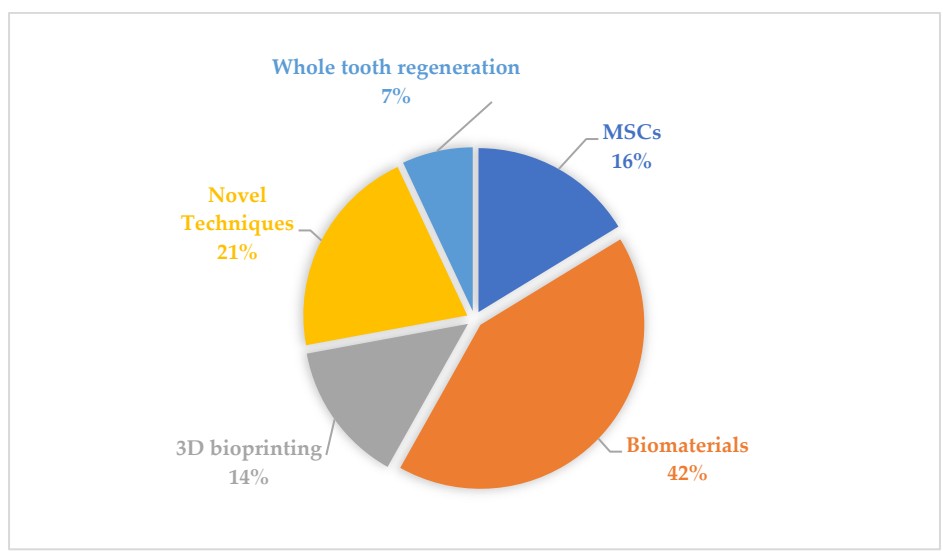

**Figure 2.** Areas of main interest of studied articles.

The columnar graph (Figure 1) shows the number of searched articles in a particular year, and represents an increasing or decreasing trend of articles' publication for the relevant subsection group of covered topics throughout the years.

The pie chart represents the individual distribution of the objects of interest of each subsection in the scoped articles. As far as the main objective of this narrative review was to explore current trends and progress in regenerative dentistry, the top 10 cited articles were gathered according to their current number of citations within the field.

### 2.2. Evaluation of the Most Commonly Used Materials

To assess the most commonly used materials in scaffold manufacturing during the last 10 years, the research methodology based on Artificial Intelligence (AI), particularly large language models (LLMs), was employed. This AI research was conducted on 20 December 2023. A large language model (LLM) is a deep learning algorithm that can perform a variety of natural language processing tasks. Large language models use transformer models and are trained using massive datasets. These can be implemented as assisting analytical systems [30] or support systems in modern dental education [31]. The version of the large language model used was Bard v2.0, released on 20 December 2023, based on the Gemini neural network architecture. This new version of the large language model, Bard, was utilized to investigate the evolution of materials employed in scaffold production for regenerative dentistry over the past decade. The prompt provided to Bard was:

*"What materials were used in scaffold manufacturing in regenerative dentistry during last ten years? Provide a timeline with highlighting the most popular/researched material of each particular year."*

The model was instructed to generate a comprehensive timeline encompassing the most prevalent and thoroughly studied materials employed for scaffold fabrication within each of the ten years under consideration.

To ensure the validity of the AI-generated timeline, it was compared against the outputs of another large language model, ChatGPT v. 3.5. This cross-verification process enabled the identification and resolution of any discrepancies or inconsistencies between the two models' responses.

Subsequently, the findings obtained from the AI analyses were subjected to expert validation by a panel of researchers specializing in scaffold manufacturing for regenerative dentistry. This expert review aimed to assess the accuracy, comprehensiveness, and relevance of the AI-generated timeline and list of materials.

### 3. Results

The number and composition of articles identified by the search query are depicted in Figures 1 and 2 and Table 2. The highest number of searched articles was published in 2021 ($n = 12$), whereas the smallest number of searched articles was published in 2022 ($n = 5$). Based on the main topic of the articles, a division into five groups was made, showing that overall, biomaterials were discussed the most (42 per cent of all articles). Interestingly enough, 80 per cent of all articles published in 2022 studied biomaterials.

The results of AI-human evaluation of the most commonly employed materials used for scaffold fabrication in the last 10 years from an AI-generated list with a timeline is shown in Table 3. Findings obtained from the AI analyses were subjected to expert validation by a panel of researchers specializing in scaffold manufacturing for regenerative dentistry. These researchers then identified the most current resources for each of the listed materials. This review process aimed to ensure the accuracy, comprehensiveness, and relevance of the AI-generated timeline and list of materials.

**Table 2.** Top 10 most cited articles relevant to the search query.

| # | Authors | Title | Citations | Reference | Published |
|---|---------|-------|-----------|-----------|-----------|
| 1 | Tahriri et al. | Graphene and its derivatives: Opportunities and challenges in dentistry. | 147 | [32] | 2019 |
| 2 | Tatullo et al. | PLA-Based Mineral-Doped Scaffolds Seeded with Human Periapical Cyst-Derived MSCs: A Promising Tool for Regenerative Healing in Dentistry. | 71 | [33] | 2019 |
| 3 | Ducret et al. | Design and characterization of a chitosan-enriched fibrin hydrogel for human dental pulp regeneration. | 54 | [34] | 2019 |
| 4 | Matichescu et al. | Advanced Biomaterials and Techniques for Oral Tissue Engineering and Regeneration-A Review. | 50 | [35] | 2020 |
| 5 | Ma et al. | Three-dimensional printing biotechnology for the regeneration of the tooth and tooth-supporting tissues. | 44 | [36] | 2019 |
| 6 | Yelick et al. | Tooth Bioengineering and Regenerative Dentistry. | 43 | [37] | 2019 |
| 7 | Prahasanti et al. | Exfoliated Human Deciduous Tooth Stem Cells Incorporating Carbonate Apatite Scaffold Enhance BMP-2, BMP-7 and Attenuate MMP-8 Expression During Initial Alveolar Bone Remodeling in Wistar Rats (Rattus norvegicus). | 38 | [38] | 2020 |
| 8 | Alipour et al. | The osteogenic differentiation of human dental pulp stem cells in alginate-gelatin/Nano-hydroxyapatite microcapsules. | 36 | [39] | 2021 |
| 9 | Sukpaita et al. | Chitosan-Based Scaffold for Mineralized Tissues Regeneration. | 34 | [40] | 2021 |
| 10 | Baranova et al. | Tooth Formation: Are the Hardest Tissues of Human Body Hard to Regenerate? | 31 | [41] | 2020 |

**Table 3.** Overview of the most used materials in "hard tissue" regenerative dentistry.

| Material | Type | Advantage | Disadvantages | Cit. |
|----------|------|-----------|---------------|------|
| Collagen | Organic | Collagen is one of the most frequently used biopolymers in the preparation of scaffolds for the regeneration of hard tissues of the oral cavity. It acts as the fundamental biological component for various tissues in the oral and craniofacial area. Its minimal immunogenicity, excellent biocompatibility, and straightforward preparation methods from diverse sources make collagen a favorable choice as a potential commercial ingredient for creating biomaterials. Collagen can be effectively modified by many chemical and physical approaches to fabricate scaffolds in different forms (e.g., membranes, sponges, gels). Furthermore, incorporating inorganic elements like hydroxyapatite (HAp) and β-tricalcium phosphate (β-TCP) through hybridization can result in the development of mineralized collagen scaffolds. This enhances the scaffolds' mechanical properties, biodegradability, and ability to induce osteogenesis. | Collagen scaffolds derived from natural sources through freeze-drying or electrospinning exhibit insufficient mechanical strength and biostability. This inadequacy has prompted persistent endeavors to enhance these scaffolds through physical, chemical, and biological modifications. | [42–45] |
| Gelatin | Organic | Gelatin, as a hydrophilic polymer, exhibits exceptional sol–gel transition characteristics and biocompatibility, rendering it a versatile material within the realm of hydrogels. Utilizing gelatin as a matrix for hydrogels enables the replication of diverse tissue characteristics and facilitates the customization of hydrogel properties, including mechanics and degradation. This adaptability makes it well-suited for a broad spectrum of biomedical applications. Studies have shown that the dental light-curing process of gelatin can sustain the viability of adult dentin cells, highlighting its potential application in the field of dentistry. Furthermore, experiments conducted in vitro demonstrated the noteworthy bioactivity of the hydrogels, as they effectively preserved the chondrocyte phenotype while fostering cell adhesion and proliferation. | Gelatin is characterized by insufficient mechanical strength. It is not suitable for applications that demand advanced adjustability in terms of cell adhesion, migration, and degradation mediated by cells. | [46–48] |

**Table 3.** *Cont.*

| Material | Type | Advantage | Disadvantages | Cit. |
|---|---|---|---|---|
| Chitosan | Organic | Chitosan, a natural biomaterial primarily derived from chitin, possesses several advantageous characteristics, including biocompatibility, hydrophilicity, biodegradability, and a wide-ranging antibacterial spectrum that encompasses both Gram-negative and Gram-positive bacteria, as well as fungi. Furthermore, its molecular structure features reactive functional groups, offering numerous sites for reactions and opportunities to establish electrochemical connections at the cellular and molecular levels. Chitosan support cell proliferation and cellular activity of osteoblasts and chondrocytes. In addition, research efforts have extensively explored composite formulations involving chitosan and hydroxyapatite, aiming to create templates of chitosan and hydroxyapatite through innovative methodologies. | Chitosan's limitations in the regeneration of hard tissues in the oral cavity include challenges such as its mechanical properties, potential degradation issues, and the need for further research to optimize its effectiveness in this specific application. | [49–52] |
| Polylactic-co-glycolic acid (PLGA) | Organic polymer | PLGA is generally considered to be a biocompatible material, meaning that it is well-tolerated by the body. PLGA is a biodegradable material, meaning that it breaks down over time into naturally occurring metabolites. This property makes it suitable for applications where the material needs to be eliminated from the body over time. PLGA has good mechanical properties, making it suitable for a wide range of applications. For example, PLGA is used to make surgical sutures that need to be strong enough to hold a wound together, but also flexible enough to not break.The rate of degradation of PLGA depends on the ratio of L-lactic acid to glycolic acid in the copolymer. Copolymers with a higher content of L-lactic acid degrade more slowly than copolymers with a higher content of glycolic acid. This property can be an advantage in some applications, such as the production of implants that need to last for a long time. | The degradation rate of PLGA depends on the ratio of L-lactic acid to glycolic acid in the copolymer. Copolymers with a higher content of L-lactic acid degrade more slowly than copolymers with a higher content of glycolic acid. This property can be a disadvantage in some applications, such as the production of implants that need to last for a long time. PLGA is more expensive than some other materials used in medicine. In some cases, PLGA toxicity can occur, usually caused by the L-lactic acid monomer. PLGA toxicity can be particularly problematic in applications where the material is in contact with blood or other body fluids. In some cases, allergic reactions to PLGA can occur. These reactions are usually mild and go away on their own, but sometimes can be severe and even fatal. | [53–56] |
| Polycaprolactone (PCL) | Organic polymer | PCL is generally considered to be a biocompatible material, meaning that it is well-tolerated by the body. PCL also promotes a very good biodegradability, meaning that it breaks down over time into the natural metabolites. This is an advantage for applications where material needs to be eliminated from the body over time. PCL has good mechanical properties, making it suitable for a wide range of applications.: PCL is easily moldable and processable, making it easy to use in medical applications. | The rate of degradation of PCL depends on the ratio of caprolactone to other monomers used in its production. Copolymers with a higher content of caprolactone degrade more slowly than copolymers with a lower content of caprolactone. This property can be a problem in some applications, such as implants that need to last for a long time. PCL is more expensive than some other materials used in medicine. In some cases, allergic reactions to PCL can occur. | [53,57,58] |
| Alginates | Organic | Alginates are generally considered to be biocompatible materials, meaning that they are well-tolerated by the body. This property makes them suitable for applications where the material needs to be in contact with human tissue. They are also biodegradable, and can break down into naturally occurring metabolites. This property makes them suitable for applications where the material needs to be eliminated from the body over time.Alginates have good mechanical properties, making them suitable for applications where the material needs to be strong enough to perform its function. Alginates also presents good liquid absorption, making them suitable for application where material needs to absorb fluids from the body. Alginates have antibacterial properties, making them suitable for applications where it is necessary to prevent infection. | The degradation rate of alginates depends on the ratio of mannuronic acid to guluronic acid in the polysaccharide. Alginates with a higher content of mannuronic acid degrade faster than alginates with a higher content of guluronic acid. This property can be a disadvantage in some applications, such as the production of implants that need to last for a long time. Alginates are more expensive than some other materials used in medicine. In some cases, allergic reactions to alginates can occur. | [59–61] |

**Table 3.** *Cont.*

| Material | Type | Advantage | Disadvantages | Cit. |
|---|---|---|---|---|
| Hyaluronic acid (HA) | Organic | HA is a linear, hydrophilic, polyanionic polysaccharide, and is a natural biological component of living organisms. It has good bioactivity, biocompatibility, and biodegradability, in the human body. The HA has multiple physiological roles, including water regulation in tissue matrices, skin wound regeneration processes, cartilage resistance to compression, act as joint lubricant and shock absorber, etc. For regenerative medicine, HA can be used as a reservoir of stimulants such as growth factors, etc. | HA properties are affected by structural and chemical complexity depending on its molecular weight, it has low mechanical strength, and may induce immunoreactivity, e.g., granulomatous foreign body reaction. | [62–68] |
| Bioactive glasses | Inorganic | Bioactive glass has potential for dental applications, such as dentin regeneration, due to its excellent bioactivity, and easy enhancement of functionality by specific therapeutic ions doping with, e.g., antibacterial and angiogenetic behavior. It has an excellent ability to bond with both hard and soft tissues. | Limited applications for low level loading replacements due to its low mechanical strength and brittleness. The processing challenges and the costs, in certain cases also slow degradation may be an issue. | [69–73] |
| hydroxyapatite (HAp) | Inorganic | HAp is a natural component of human bones and teeth. It has excellent biocompatibility and can provide stimuli for osteoinductivity and osteoconductivity. Is often used in dental applications due to its similarity to the mineral composition of natural teeth, and integrates well with the surrounding tissue. | HAp is brittle and has very low fracture toughness. Its application is complicated with difficulty in shaping. Pure HAp may have poor adhesion to soft tissues and slow integration or resorption rates. The cost of medical grade HAp are high. | [19,74–78] |
| Tricalcium phosphate (TCP) | Inorganic | Unlike Hap, the β-TCP bioceramics show higher solubility and biodegradation rate by osteoclast cells, which provoke a local acidification that leads to material dissolution. Osteoclasts then initiate bone resorption by releasing protons and enzymes. This process of bone resorption caused by osteoclasts is coupled with ossification of osteoblasts. By testing β-TCP ceramics, they proved their ability to support differentiation and proliferation of osteoblasts and mesenchymal cells. It has been reported to have excellent biocompatibility and osteoconductivity as well. | Lower mechanical strength as HAp ceramics. | [79–83] |
| Biphasic calcium phosphate (BCP) | Inorganic | Biphasic calcium phosphate has been developed as a compromise to get good mechanical properties of HAp and higher solubility and osteoconductivity of β-TCP. It is considered the gold standard of bone substitutes in bone reconstructive surgery. The advantage of BCP is the preservation of the mechanical strength during its resorption. The higher the ratio, the greater the resorbability. BCP-A (contains high amount of Calcium-deficient hydroxyapatite CDHA) significantly decreased the inflammation response of dental pulp and promotes the formation of dentin bridges. The BCP with composition of 15% HAp and 85% β-TCP forms the bone earlier and in more quantity than second-investigated BCP with composition of 85% HAp and 15% β-TCP in mandible bone of beagle dogs after 4, 12, and 26 weeks. | The ratio β-TCP/HAp should be individually tuned according to application (depending on the solubility—increased solubility of bio ceramics does not mean that resorption activity is optimal). | [84–89] |
| Calcium Phosphate Cements (CPCs): | InorganicTypical CPCs [1] | Paste—set in situ to fill bone defects. CPCs have shown potential in dental applications for filling cavities, repairing defects, and promoting bone regeneration. The main characteristic and advantage of CPCs is their injectability and/or moldability to fill optimally irregular bone defects. They form intimate contact with the bone structure ensuring good transformation into new bone. The mechanical properties, as well as setting time, varies depending on the chemical composition of CPCs. New kinds of CPCs can reach high compressive strength up to 35 MPa and setting time of 14 min. | The dense structure of CPCs lacking the microporosity that is necessary for bone ingrowth together with the slow biodegradation represent their main disadvantages. | [90–95] |

[1] Consists of a calcium phosphate-based powder and a liquid component, which, upon mixing, undergo a non-toxic chemical reaction, resulting in setting and hardening.

## 4. Discussion

This article is the first narrative review characterizing scaffolds and clinical approaches in regenerative dentistry over the last five years (2019–August 2023). The review aims to elucidate current context and options in the field of regenerative dentistry, and investigates possible research directions.

### 4.1. Mesenchymal Stem Cells and Their Application in Regenerative Dentistry

Mesenchymal stem cells (MSCs) are one of the elements which are crucial to the process of tissue regeneration [64,96–98]. They can be, in their entirety, quite easily extracted from various types of body tissues; however, they strictly require suitable conditions to fully prosper. Different types of MSCs exert various effects on different kinds of human tissues and numerous studies and clinical applications have confirmed utility and advantageous properties of mesenchymal stem cells as a part of scaffold grafts in regenerative dentistry [99–102].

Bone marrow stem cells (BMSCs) were seeded onto collagen scaffolds that have shown extracellular matrix (ECM) mimicking ability, which is crucial for biological processes and cell synthesis. Additionally, vascularization and new bone tissue formation were boosted via an increase in BMP-2 and b-FGF markers [43]. Dental stem cells (DSCs) can be retrieved from several parts of the oral cavity, e.g., from deciduous teeth, apical papillae, and periapical cysts. Apart from DSCs, adipose tissue stem cells (ADSCs), induced pluripotent stem cells (iPSCS) and exosomes, which are small membrane vesicles isolated from cells, are of greatest importance in regenerative dentistry [33,103,104].

Many authors have studied applications and effectiveness of MSCs in relation to various scaffold media. Stem cells derived from exfoliated human deciduous teeth (SHEDs) were studied by Prahasanti et al. SHEDs were seeded onto carbonate-apatite scaffolds to support the bone remodeling process in post-extraction cavities. The study in vivo showed higher levels of bone morphogenic proteins (BMP-2 and BMP-7) and a lower degree of matrix metalloproteinase (MMP-8) expression together with better osteogenesis in rats [38].

In the study by Ha et al., the impact of material stiffness and topographical micropatterned alignment of Gel-MA hydrogels on apical papilla stem cells (APSCs) was explored. Cells expressed higher viability, higher proliferation response, guided self-alignment, as well as an increase in ALP expression, leading to better odontogenic differentiation ability compared to scaffolds with no micropatterned structural alignment [105].

MSCs derived from human inflamed periapical cysts (hPCy-MSCs) seeded onto PLA-mineral doped (dicalcium phosphate dihydrate (DCPD), and/or hydraulic calcium silicate (CaSi)) scaffolds promoted higher DMP-1 and RUNX-2 activity, resulting in better osteogenesis and cell proliferation rates [33]. Kanjevac et al. studied a short peptide sequence with the potential to activate and inhibit osteogenesis [104].

As an alternative for regular stem cells, Ana et al. presented how exosomes participate in cellular regulatory processes. Interestingly, their potential rises when combined with suitable ceramic scaffolds [74,106]. There is a need to further investigate mechanisms of MSCs' action, integration within scaffold systems, their responses to multiple factors.

### 4.2. Materials for Scaffold Fabrication in Regenerative Dentistry

Scaffolds are generally considered as supporting pillars for stem cells that are to be grown. They can be fabricated from different materials, including biomaterials or their combinations. Material selection is crucial for proper in vivo functioning.

A significant progress in scaffold fabrication from biocompatible materials has been made over the past years, presenting properties that allow them to become alternatives for conventional grafts. The selection of materials (including nanomaterials) [72,107,108] determines properties of scaffolds. The aim usually is to get as close as possible to native tissue characteristics. Scaffolds themselves influence the properties, behavior, growth, and functioning of stem cells seeded within [109].

The range of use of biomaterials in regenerative dentistry keeps increasing. Biomaterials can enhance cell proliferation rates, viability. They can modify responses of cells, increase their surface adhesion, differentiation, osteoblastic and odontoblastic activity, osteoconductivity, mineralization processes, antimicrobial effects, vascularization, and other mechanical and biological functions. These numerous effects determine the applications of biomaterials in regenerative dentistry [110–113]. In fact, biomaterials in the form of scaffolds have shown favorable properties in restorative dentistry, endodontics, implantology, and maxillofacial surgery [110,114,115].

Graphene-based scaffolds offer biocompatibility, anti-bacterial properties, and the ability to stimulate cellular processes. This makes them promising for tissue engineering, though further research on potential cytotoxicity is needed [32,116].

The applications of chitosan were also explored in several studies [34,40,111,113,117]. It was concluded that chitosan itself has poor mechanical properties, undergoes rapid degradation, and has limited use due to rather low osteoconductivity [40,118,119]. Therefore, chitosan and graphene oxide were combined with hydroxyapatite and xanthan gum to form Chitosan/Xanthan/Hydroxyapatite–Graphene Oxide scaffold, which has enhanced biological properties and high cell viability in a MTT cytotoxicity test [117].

Sato et al., combined chitosan nanofibers with nano-hydroxyapatite particles, resulting in high antibacterial and antibiofilm effects against *P. gingivalis* [111].

Anastasiou et al. discovered that $Ce^{3+}$-doped fluorapatite integrated into chitosan scaffolds increases osteoconductivity levels, enhances osteogenic differentiation, and boosts antibacterial activity more compared to the $Sr^{2+}$-doped fluorapatite–chitosan scaffold when assessed with dental pulp stem cells (DPSCs) [113].

DPSCs also improved their cell proliferation rates when gelatine as a co-polymer was added into the chitosan-based scaffold [120]. Other researchers developed fibrin-based hydrogels supplemented with chitosan nanoparticles for dental pulp regeneration purposes, which also showed good antimicrobial activity against highly resistant *E. fæcalis* often located in infected root canals [121–124]. In addition, fibrin-chitosan hydrogel did not have negative effects on dental pulp cell viability, proliferation, morphology, and collagen production [34].

Furthermore, the study from Thurzo et al. explored gyroid and rectilinear 3D-printed hydroxyapatite scaffolds made from novel composite filament, which were sintered at 1300 °C and 1400 °C with high biocompatibility and in vitro cell adhesion [112].

Lin et al. presented alginate as a material with high potential in regenerative endodontics due to its wide spectrum of applications. Besides being used in a form of scaffold, alginate can be a cell-carrier, a microcapsule delivery agent, an agent used for boosting chelating properties of hypochlorite, and a root canal sealer [125].

On top of that, Naik highlighted the importance of 3D-printed scaffolds, including cells and signaling molecule selection, in promoting increase in root dimensions, which greatly reduced incidence of tooth fractures in necrotic immature permanent teeth. He calls for an innovation of endodontic treatment protocols, shifting towards regenerative apexification techniques [115].

Another element that is used in regenerative dentistry is porous boron-modified bioactive glass that possesses dentin regenerative properties, while having good biodegradation and inducing calcium phosphate formation [72]. Another promising actor in the studied area is baghdadite I—calcium silicate reinforced with zirconium ions. It increases apatite formation, mineralization, and ossification, which plays a crucial role in dentofacial tissue regeneration [110].

Gelatine methacrylate (Gel-MA) is another biomaterial used in tissue engineering, drug delivery systems, and 3D bioprinting applications. An example of such an application would be fabrication of patient-specific hydrogels with tissue healing properties and controlled release of bioactive molecules or substances [126].

In relation to drugs-doped scaffolds, Soares et al. described simvastatin (SV) as a promising agent in odontoblastic marker overexpression and matrix mineralization

enhancement when in contact with dentin [127]. Another study explored deproteinized bovine bone minerals (DBBM) in the form of granules, and blocks and their effect on osteoblasts and macrophages, showing promising outcomes—even more pronounced in the case of DBBM granules [128].

Yu et al. compared mineralization rate, properties, and indications of cell-free biomimetic mineralization and cell-dependent scaffold mineralization in skeletal and dental hard tissue regeneration [129]. Biomimetic principles are significantly contributing to advances in this field [130,131].

Further research on materials for regenerative dentistry is crucial, especially regarding their structure, degradation, biocompatibility, stem cell interaction, vascularization, and antimicrobial properties.

### 4.3. 3D Bioprinting in Regenerative Dentistry

Bioprinting as a process plays an important role in tissue regeneration. The selection of materials (including bioinks), form, methods, and design of the future product are some of the elements which need to be considered to obtain scaffolds with the potential to replace living tissues [132,133].

The main goal is to approximate the anatomy of living tissues as precisely as possible [134]. Bioprinting nowadays can provide different components for the whole construction, thus helping substitute conventional treatment methods [8]. 3D bioprinting finds its place in the complex regeneration of teeth and its structures, including bioprinting of cells, matrix materials, and thus controlling the external and internal properties of regenerated modules [36]. Various fabrication approaches have been explored in different fields of regenerative dentistry, including the use of hydrogels, 3D scaffolds, and thin films. For example, skin tissue substitutes and wound dressings have been used in the maxillofacial region [135]. Bioink development is also very crucial for scaffold bioprinting. A study by Mohabatpour et al. presented novel bioink, containing dental epithelial cells in an Alginate-Carboxymethyl Chitosan mixture for enamel tissue regeneration with good printability, porosity, integration, and high cell differentiation ability and viability [136].

Bioreactors also take part in the bioprinting strategy selection. Several types of bioreactors can be used, such as flow perfusion, spinner flask, or the rotational ones. Their purpose is to mimic in vitro conditions for scaffolds and stem cells, modulate their behavior, and assess their biological properties [137].

Bioprinting success relies on careful material selection, biological studies, and cell choices. It holds promise for revolutionizing regenerative dentistry and tissue engineering [138]. Further clinical studies are needed to determine the optimal filaments, bioinks, and bioreactors for 3D bioprinting of teeth structures.

### 4.4. Novel Techniques and Modifications in Scaffold Fabrication

The selection of the scaffold design, stem cells, growth factors and environmental conditions suitable for vascularization are key aspects for successful tissue regeneration, which Osypko et al. termed the Diamond concept of healing [139]. Several modifications have shown great potential to enhance the scaffold properties, as well as the growth, of stem cells. Modifications are conducted in different ways.

Nanotopography, decellularization, melt-electrowriting technique, or microencapsulation of stem cells are all processes which impact surface structure of scaffolds, their mechanical strength, cell proliferation, osteogenic properties, degradation, and vascularization of scaffolds. So far, however, nanotopography in the dental pulp complex regeneration is not yet fully investigated [39,140–142].

Nopal scaffold decellularization seems to be a very promising technique in regenerative dentistry. Further research is needed, especially in computational simulation [NO_PRINTED_FORM], in silico approaches [NO_PRINTED_FORM], and contact pressure 3D models [NO_PRINTED_FORM] in order to study the performance of nopal scaffolds effectively and predict the outcomes with higher level of certainty [141,143–145].

The foam replication method is one of the methods to create novel scaffolds for bone augmentation. Fabricky et al. (2021) compared the traditionally used Cerabone and novel scaffolds, produced by the aforementioned method, to outline future research directions [146].

The osteogenic potential is a very important factor in achieving complex and functional construction of cell-seeded scaffolds. The technology that uses 3D-printed molds to create teeth-shaped fibrin gel implants may be very helpful for further exploring and testing [147].

Scaffold-based techniques have shown very positive results in bone and other tissue regeneration processes. On the other hand, studies on scaffold-free techniques are scarce. Tatullo et al. evaluated scaffold-free techniques (an approach using exosomes, hypoxia-based MSCs, strategic use of heat-shock proteins) in single repair processes, and investigated their potential in regenerating smaller lesions. Dissanayaka used scaffold-free sheets and spheroids for the complex regeneration of dental pulp [148]. On the other hand, Tatullo et al. also highlighted and confirmed higher potential and better functionality of scaffold-based techniques in the repair of large, damaged lesions when compared to scaffold-free techniques [149].

Several novel techniques have recently been enabled by the current significant advances in technologies, such as optical scanning [150], AI segmentation [131,151], and micro-CT methods [152], which are potentiated by advances in material analytical methods, such as attenuated total internal reflection coupled with Fourier transform infrared spectroscopy, X-ray fluorescence, and many others.

*4.5. Whole Tooth Regeneration*

Whole tooth regeneration is getting much more attention now than ever before. By inventing a bioengineered tooth, which can replace all required parts of the damaged area, including periodontal ligaments, bone, cement, and the tooth itself, we could get to the point where conventional methods of tooth replacement would change. In so doing, we could overcome all disadvantages and limitations of classic implants while improving overall dental health [35,153,154].

Yelick described bioengineered roots fabrication, which he later fitted with a custom-3D-printed dental crown, while being more affordable and achievable than conventional treatment methods [37]. Understanding signal pathways of dental tissue genesis, development of the scaffolds and relevant drug release systems, as well as utilization of adult stem cells, iPSCs and tooth germ cells can help reveal directions that should be pursued within the domain of teeth bio-manufacturing research [41].

Successful biological tooth regeneration could be one of the first steps in organ regeneration. Whole tooth engineering or regeneration remains very complicated, and highlights several problems, such as programming the stem cells to differentiate into tooth-specific cell types.

When it comes to whole tooth regeneration, one needs to be critical about contemporary papers in other research areas as well. Seemingly, the whole concept of scaffolds used as backbones in fabricating biocompatible tissues, potentially filled with living cells that integrate within organisms, is quite complex and faces many challenges. Despite many advances and much success in many of the research directions, there is a different way of thinking when it comes to whole tooth regeneration needs. Katsu Takahashi, as the leader of research from Medical Research Institute Kitano Hospital in Osaka, has been focusing on tooth regrowth development research since 2005 at Kyoto University. Takahashi and his research team identified the USAG-1 gene, which led to the development of a neutralizing antibody medicine capable of blocking the protein's function. Successful animal experiments, where mice were treated with the medicine grew additional teeth, underscore the potential of this innovative approach [155].

The study by Takahashi et al. states that many genes responsible for congenital tooth agenesis have been identified, and many are common in humans and mice. As an example, the RUNX-2 causative gene for congenital tooth agenesis is mentioned in

the study. Suppression of the function of this gene leads to arrested development of the tooth. In mice models, tooth arrested development was rescued via double-knockout of RUNX-2 and USAG-1 [156]. This suggest that targeted molecular therapy could generate teeth by stimulating arrested tooth germs in patients with congenital tooth agenesis [157]. Takahashi's research also holds promise by enabling the growth of new teeth among children suffering from anodontia from a very young age [158,159].

The revolutionary step comes with the latent ability of human beings to grow the third set of teeth, which was determined on certain animals [159,160]. As the team prepares for clinical trials in 2024, global anticipation and attention surrounding this tooth regrowth medicine continue to grow. With the aim of making it available for general use by 2030, Takahashi and his team stand at the threshold of a new era in dentistry where tooth regeneration becomes a reality, reshaping our approach to oral health and dental care.

Clinical approaches and other studies need to be done to fully understand the concept of whole tooth regeneration [37,161]. Finally, financial investments need to be made to advance hereinabove mentioned techniques to the mainstream within regenerative dentistry.

### 4.6. Future of Scaffold Approaches

Future trends in scaffold approaches show promising opportunities for tissue regeneration in the near future. Early explorations of using 4D materials in scaffold development [162] or gene-activated materials [163] or various advanced scaffolds for dentin–pulp complex regeneration and other innovative trends in regenerative dentistry [6,109,110,164–168] suggest an incoming boom of this specialty.

Novel materials that will be used in scaffold approaches include decellularized matrices, hydrogels, scaffolds, and bioactive materials. The fabrication methods are based on 3D printing, self-assembling microfabrication, electromagnetic patterning, and bioprinting, complemented by using growth factors and other molecules. These emerging technologies have the potential to revolutionize scaffold approaches and lead to the development of new and more effective therapies for a wide range of problems.

The following trends were identified according to the literature analysis:

- Decellularized matrices are natural scaffolds created by removing cells from tissue. They are biocompatible, biodegradable, and can be tailored for specific tissue regeneration, such as using heart matrices to regenerate heart muscle [141].
- Hydrogels are soft, flexible materials made from natural (e.g., collagen) or synthetic polymers. Their customizable properties like stiffness, degradation, and cell adhesion make them versatile for supporting a wide range of tissue growth.
- 3D printing is a technology that can be used to create complex scaffolds with intricate structures that mimic the natural extracellular matrix (ECM) of tissues. This can help to improve the ability of scaffolds to support cell growth and differentiation. 3D printing is also a relatively rapid and efficient process, which can make it a more cost-effective way to produce scaffolds or even personalized medical appliances [151,169].
- Adding growth factors and other molecules to scaffolds improves their performance in tissue regeneration by promoting cell growth, differentiation, and tissue formation.
- Self-assembling scaffolds are materials that can spontaneously assemble into complex structures without the need for external forces. This can lead to the formation of scaffolds that are highly porous and interconnected, which is ideal for supporting cell growth and tissue regeneration.
- Bioactive materials are materials that can release bioactive molecules, such as growth factors and signaling molecules, over time. This can help to promote cell growth, differentiation, and tissue formation. Bioactive materials can also be used to deliver drugs and other therapeutic agents to cells and tissues.
- Microfabrication is a technology that can be used to create scaffolds with micrometer-scale features. This can be used to control the size and shape of pores in scaffolds, which can affect the ability of cells to adhere and grow on the scaffold.

- Electromagnetic patterning is a technology that can be used to create scaffolds with patterns of electrical charges. This can be used to attract and guide cells to specific locations on the scaffold.
- Bioprinting is a technology that can be used to create scaffolds with complex structures using living cells. This can be used to create scaffolds that are more similar to natural tissues and that can support the growth of a wider variety of cell types.

## 5. Conclusions

### 5.1. Overall Conclusions

1. Multi-Material Scaffolds are Key: The most impactful research emphasizes the need for combining various materials in biocompatible scaffolds to achieve tailored properties and optimal biological responses in hard tissue regeneration.
2. Focus on Stem Cell Interaction: Studies with the highest impact explore how scaffold materials influence stem cell proliferation, differentiation, and behavior. Understanding these material-cell interactions is crucial for developing successful therapies.
3. Novel Modifications are Promising: Advancements in nanotechnology, 3D bioprinting, and surface modification techniques have the potential to revolutionize scaffold design, increasing their efficiency and customization for regenerative dentistry.

### 5.2. Specific Conclusions

1. Graphene, Chitosan, and Composites: Graphene demonstrates antibacterial properties and cellular stimulation, making it a valuable candidate. Chitosan, while needing improvement on its own, shows promise when combined with other materials like hydroxyapatite.
2. Bioprinting for Tailored Solutions: 3D bioprinting shows tremendous promise for creating patient-specific scaffolds, driving greater customization and success rates in dental tissue regeneration. This includes bioprinting of cells, matrix materials, and entire tooth structures.
3. Importance of Cell Source: Exploration of mesenchymal stem cells (MSCs) from different sources (dental pulp, bone marrow, adipose tissue) in conjunction with scaffolds is highly significant for determining optimal cell-material pairings for specific applications.
4. Newer Materials Emerge: Bioactive glasses, boron-doped biomaterials, and unique composites hold promise for enhanced bone and tooth regeneration.

### 5.3. Future Directions

- Biomimetic Approaches: Further emphasis on biomimetic principles, mimicking natural tissue structures and compositions, will likely drive future scaffold material and design innovations.
- Clinical Translation: A strong need exists to translate promising laboratory findings on scaffold-based materials and approaches into clinical dentistry, paving the way for more effective and available treatments.
- In-depth Material Investigations: Continued in-depth research on biocompatibility, degradation rates, cell interactions, and potential cytotoxicity of novel and complex scaffold materials is essential.
- Standardization: As the field matures, standardization of protocols, evaluation metrics, and reporting methods becomes critical for comparing research findings and accelerating clinical adoption.

**Author Contributions:** Conceptualization, A.T., J.T. and M.G.; methodology, A.T.; software, A.T.; validation, A.T., J.T. and M.G.; formal analysis, A.T.; investigation, M.G., P.J., J.T., Ľ.D., M.J., Ľ.B., P.P. and A.T.; resources, A.T.; data curation, A.T.; writing—original draft preparation, M.G., P.J., J.T., B.M., Ľ.D., M.J., Ľ.B., P.P. and A.T.; writing—review and editing, M.G., P.J., J.T., B.M., D.H., S.T., Ľ.D., M.J., Ľ.B., P.P. and A.T.; visualization, M.G.; supervision, A.T. and J.T.; project administration, A.T.; funding acquisition, A.T. and M.J. All authors have read and agreed to the published version of the manuscript.

**Funding:** This work was supported by the Slovak Grant Agency for Science (KEGA)—grant No. 054UK-4/2023 and by Slovak Research and Development Agency—grant No. APVV-21-0173.

**Institutional Review Board Statement:** Not applicable.

**Informed Consent Statement:** Not applicable.

**Data Availability Statement:** Not applicable.

**Acknowledgments:** The authors gratefully acknowledge the technical support of the digital dental lab infrastructure of 3Dent Medical Ltd. Company, as well as the dental clinic Sangre Azul Ltd.

**Conflicts of Interest:** The authors declare no conflicts of interest.

## Abbreviations

| | |
|---|---|
| AI | artificial intelligence |
| 3D | three dimensional |
| PLA | polylactic acid |
| PGA | polyglycolic acid |
| HAp | hydroxyapatite |
| HA | hyaluronic acid |
| TCP | tri calcium phosphate |
| LLM | large language model |
| PLGA | polylactic-co-glycolic acid |
| PCL | polycaprolactone |
| CPC | calcium phosphate cements |
| BCP | biphasic calcium phosphate |
| MSCs | mesenchymal stem cells |
| ECM | extracellular matrix |
| DSCs | dental stem cells |
| BMP-2 | bone morphogenic protein 2 |
| BMP-7 | bone morphogenic protein 7 |
| MMP-8 | matrix metalloproteinase 8 |
| b-FGF | basic-fibroblast growth factor |
| BMSCs | bone marrow stem cells |
| ADSCs | adipose tissue stem cells |
| iPSCs | induced pluripotent stem cells |
| SHEDs | human exfoliated deciduous teeth stem cells |
| APSCs | apical papilla stem cells |
| ALP | alkaline phosphatase |
| hPCy-MSCs | human periapical cyst derived mesenchymal stem cells |
| DCPD | dicalcium phosphate dihydrate |
| CaSi | calcium silicate |
| DMP-1 | dentin matrix protein-1 |
| RUNX-2 | runt-related transcription factor 2 |
| MTT | 3-(4,5-dimethylthiazol-2-yl)-2,5-diphenyl-2H-tetrazolium bromide |
| DPSCs | dental pulp stem cells |
| Gel-MA | gelatine methacrylate |
| SV | simvastin |
| DBBM | deproteinized bovine bone mineral |
| Micro-CT | micro computer tomography |
| USAG-1 | uterine sensitization-associated gene-1 |
| 4D | four dimensional |
| PEG | polyethylene glycol |

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
