# Peer review of "Evolving Strategies and Materials for Scaffold Development in Regenerative Dentistry"

_applsci, doi:10.3390/app14062270_

Round 1

Reviewer 1 Report

Comments and Suggestions for Authors

The topic of the review is focused on progress and further trends of scaffold application in regenerative dentistry. The authors overviewed the current state of the art on biomaterial utilization in dentistry from the last five years in a synthetic way and outlined the future directions of biomaterial application in dental implantology. The proposed review article addresses important issues related to modern regenerative dentistry. However, the form of the article proposed by the authors may cause some difficulties in the readers' reception of the presented content. However, in my opinion, this manuscript can be accepted after the authors consider the following comments:

1.        The text in the columns and rows in Table 3 overlaps, which does not allow you to determine exactly where a new row/column begins. Leaving more space between columns/rows could solve the problem.

2.        Table 3 also lacks a uniform style in describing the advantages of a particular type of material. Unifying the style of Table 3 is necessary to make it easier for the reader to perceive it.

3.        Another problem is the entire chapter 4. Discussion. It is 5 pages of solid text, not even separated by paragraphs. Such a presentation of chapter 4 causes difficulty in its reception by the reader. Therefore, I suggest that the authors add paragraphs in each subchapter of Chapter 4. In addition, synthesizing the presented data in each subchapter in the form of diagrams or tables would certainly facilitate readers' reception of the article.

4.        The manuscript contains many abbreviations, so a list of abbreviations should be added at the beginning of the article to help the reader find the meaning of each abbreviation.

Author Response

Response to Reviewer 1

Dear Reviewer,

Thank you for your valuable feedback. We agree that the formatting and stylistic inconsistencies in Table 3 need improvement. We have taken your suggestions into account and made the following revisions:

  • Spacing: We have increased the spacing between columns and rows in Table 3 for improved readability. This will ensure clear delineation between data points.
  • Style: We have carefully reviewed and edited the descriptions of material advantages in Table 3. They now follow a consistent structure and focus on key benefits for ease of understanding.

I believe these changes significantly improve the clarity and presentation of Table 3. I appreciate your detailed suggestions, and I am committed to ensuring the quality of the manuscript.

AD Comment 3:

Thank you for highlighting the readability issues in Chapter 4 (Discussion). We agree that the dense text block is difficult to process. We have restructured the whole chapter to incorporate paragraphs and, where possible, and have rewritten some parts.

AD Comment 4:

The list of abbreviations will be part of the article; however, its precise placement will depend on the structure of the article as directed by the editor.

We thank you for helping us improve the paper.

Kind regards,

Authors

Reviewer 2 Report

Comments and Suggestions for Authors

 1.       The idea behind the study is sound, the abstract and the introduction are well-written. Sadly, the quality of writing significantly and progressively deteriorates starting in the results section. The Authors use ChatGPT and Bard to identify important studies in the field, but later discussion analyzes completely different articles. Reading the study, I was under the impression that it was stitched together from two different articles. The Authors should make more effort to make the two parts – results and discussion more compatible. I’m sad to say that while this study contains some major achievements in the field, it completely lacks criticism, doesn’t point out research gaps, nor does it identify the main areas to debate.

Below is a list of some remarks, questions, and concerns about the study. More details are also marked by highlights + comments in the article (see the attached file).  

2.       Overall, the article lacks criticism towards the presented materials and results, and doesn’t give much outlook or the Authors’ personal opinion. Thus, it can hardly be regarded as a review and is rather a compilation of examples. The Authors are kindly referred to read about what a well-constructed review should contain (for example https://www.ncbi.nlm.nih.gov/pmc/articles/PMC3715443/). Let me just quote: Reviewing the literature is not stamp collecting. A good review does not just summarize the literature, but discusses it critically, identifies methodological problems, and points out research gaps [19]. After having read a review of the literature, a reader should have a rough idea of:

a.      the major achievements in the reviewed field,

b.      the main areas of debate, and

c.       the outstanding research question”

3.       In some cases, articles are missing from the nouns.

4.       In some cases, the Authors are using interchangeably present and past tenses – this should be unified throughout the study.

5.       Section 1.2 would benefit from specifying certain types of tissues that were aimed to be restored. Specific material types used should also be listed along with application outcomes. It would be beneficial to point out specific commercial solutions employed at different time points. The term “bioprinting” is improperly used in this section, as it means printing with cells, which is not the case here.

6.       In Table 1, column “scaffold type” could benefit from either listing the certain biomaterial used or from specifying the tissue that was aimed to be restored (preferably, both should be supplemented). Maybe converting the table into a graphic would be beneficial?

7.       I could not find the information on how exactly the Field-Weighted Citation Impact was calculated. This should be supplemented and briefly explained so that the exclusion and inclusion criteria for the articles are clear. Specific thresholds employed should also be listed.  

8.       In line 158, it should be stressed what was the article selection criteria.

9.       It is somewhat unclear to me why the Authors chose to limit the analyzed articles to mesenchymal stem cells (as is suggested in Figures 1 and 2). The term itself raises controversies (doi: https://doi.org/10.1038/d41586-018-06756-9), and the current consensus is that it is more appropriate to always describe the tissue of origin. Hence, my suggestion would be to narrow down the search to articles that work on well-defined cell types, explicitly stating the cells' origin, gene expression, etc. This would be a more qualitative than quantitative analysis, as I am aware that some of the most popular articles work on rather poorly characterized cells.

10.   There already are some more science-oriented AI large-language models (i.e., Consensus or Elicit), could the Authors explain why they chose to use more generic tools such as ChatGPT and Bard? Furthermore, the question cited on line 185 contains a grammar mistake. Could this undermine the findings?

11.   Did the AI search include review articles? From Table 2 I can see that the answer is yes. Don’t the Authors think that the conducted analysis might have been more accurate if the reviews were excluded? Reviews often contain errors and/or superficial analyses of the studies.

12.   The panel of researchers that validated the AI-generated output should be described in more detail. How many participants, what were their qualifications, how were they selected, and was there any potential bias? Were there any specific tools or methodologies used to evaluate or score the search results?

13.   In the results section, the total number of AI-analyzed articles should be listed.

14.   Table 3 is very hard to read, as its columns and rows overlap each other. Clearer spacing should be used. Further issues with this table:

a.       Perhaps this table could benefit from changing the page orientation to horizontal.

b.       Maybe some of the “advantages” could be shortened?

c.       Disadvantages could also be corrected – for example, in chitosan, the information presented is so generic, that no consensus could be reached based on it.

d.       In PLGA, the same information about the material’s rate of degradation is repeated as an advantage and disadvantage. This should be corrected. Listing sutures as an example of applications with demand for high mechanical properties seems unfortunate, especially in a review that concerns its usage to regenerate bone. L-lactic acid is a metabolite naturally occurring in the human body – it is generally not regarded as toxic, so stating the opposite seems incorrect.

e.       The same advantages and disadvantages are repeated for different materials (including some definitions) – these should be shortened.

f.        Listing prices as PLGA’s, PCL’s, and alginate’s disadvantage are not accurate – most of the cited materials are more expensive – especially, biopolymers.

g.       For PCL, PLGA, and alginate, disadvantages claim that the degradation can be steered and that this could be problematic. This makes no sense, the fact that the degradation rate can be tailored to specific needs is among the most important advantages of using synthetic polymers over biopolymers.

h.       It is not true that alginates have good mechanical properties, compared to PCL or PLGA, they are rather weak (and the same goes for collagen, gelatin, and chitosan). Alginate’s liquid absorption properties are also true for collagen, gelatin, and chitosan.

i.         Alginate and chitosan do not have intrinsic antibacterial properties – these could be granted by specific modifications.

j.         It is unclear what the Authors mean by saying that HA “can be used as a reservoir of stimulants such as growth factors, etc.

k.       Claim that “HA properties are affected by structural and chemical complexity depending on its molecular weight” is imprecise. Structural and chemical complexity does not have to be anyhow correlated with molecular weight. Furthermore, all of the cited polymeric materials will have their properties determined by their molecular weights.

l.         What is the reason why some of the materials are listed with subsections and others – are not?

m.     Bioactive glass constitutes a group of materials with different properties. Using it as a category would make the same sense as listing all the polymers in one category as polymers.

n.       The Authors use the same abbreviation for hyaluronic acid and hydroxyapatite, which is misleading – this should be corrected.

o.       It is not true that BCP is a gold standard for bone substitution.

p.       The whole section about BCP is poorly written and needs numerous corrections. The fact that BCP composition can be tailored to meet specific needs is certainly NOT a disadvantage.

q.       Calcium phosphate cements, which are made by mixing calcium phosphate with water, should not constitute a separate category – this is rather an application form, not a material type.

r.        The table contains numerous linguistic, grammar, and punctuation mistakes.

15.   In line 252, stem cells are a broader category than mesenchymal stem cells and thus, should not be used herein.

16.   Cells cannot show “ECM-mimicking ability”, rephrasing is needed (lines 253 – 254)

17.   Section 4.1 lacks any structure – different cell types and materials are intermixed with each other. Some logical grouping should be employed. The Authors should also identify the specific application that the cited study is suggesting.

18.   Whenever in vitro studies are cited, more experimental details should be given (culture length, what and how was tested, whether it was a direct seeding or culture on extracts).

19.   Whenever in vivo studies are cited, the actual animal used should be listed.

20.   In all cases, specific applications of scaffolds should be given – dentistry encompasses the regeneration of different tissue types, and there are different requirements and challenges for each of them.

21.   In some instances, the Authors suggest regeneration of bones, in others of teeth, and in others - of skin. This is confusing and such a mixture should not be presented intermixed throughout the sections.

22.   Section 4.2 lacks any structure. The same idea is repeated multiple times, but differently from lines 283 to 297 – this should be shortened. different materials and modifications are intermixed with each other. Some logical grouping should be employed. The Authors should also identify the specific application that the cited studies are suggesting.

23.   MTT analyzes the cell metabolic activity, so claiming that it can be used to evaluate materials’ “biological activity” is a huge overstatement.

24.   The sentence in lines 285 – 286 is pure tautology as no scaffold isn’t made of biomaterials.

25.   Section 4.3 also lacks structure. Some examples of 3D printing are given, with no consensus or takeaway message.

26.   Section 4.4 seems to be stitched up from random sentences – sometimes even two different sentences are combined into one (e.g. lines 391-395). The whole section should be carefully rewritten to show a clear flow of thoughts and provide consensus and direction in which the field is evolving.

27.   Section 4.5, while being extremely interesting, hardly has anything to do with the subject of this review (scaffolds for regenerative dentistry). The studies conducted by Takahashi and his team concerned modification of the expression of certain genes to force the body to grow teeth. Hence, this relates to molecular biology and dentistry and does not involve any materials.

28.   The Authors should specify what they mean by 4D materials.

29.   A summary of trends identified in section 4.6 should focus on the subject of this review – i.e., it should concern materials used for dental tissue engineering (and not a heart or “various tissues”.

30.   The Authors claim to have identified some trends, but these were not presented in this review: self-assembly, microfabrication, electromagnetic patterning.

31.   Bioprinting is a subclass of 3D printing and should be presented in the same section

32.   A separate section of this chapter should concern chemical and biological modifications of the scaffolds.  

33.   This review could benefit from some multipanel images that represent examples of some important developments in regenerative dentistry using scaffolds.

34.   I would suggest adding a section that lists commercial applications based on the R&D advancements in the field.

Comments on the Quality of English Language

Lengthy sentences, 

Missing or obsolete commas

Improper words used

Some sentences shift the subject, giving the feeling that they are stitched together from two (or more) separate thoughts. 

More details can be found in the previous section as well as in the attached file.  

Author Response

Reviewer 2

Comments and Suggestions for Authors
 1.       The idea behind the study is sound, the abstract and the introduction are well-written. Sadly, the quality of writing significantly and progressively deteriorates starting in the results section. The Authors use ChatGPT and Bard to identify important studies in the field, but later discussion analyzes completely different articles. Reading the study, I was under the impression that it was stitched together from two different articles. The Authors should make more effort to make the two parts – results and discussion more compatible. I’m sad to say that while this study contains some major achievements in the field, it completely lacks criticism, doesn’t point out research gaps, nor does it identify the main areas to debate.
Below is a list of some remarks, questions, and concerns about the study. More details are also marked by highlights + comments in the article (see the attached file). 
2.       Overall, the article lacks criticism towards the presented materials and results, and doesn’t give much outlook or the Authors’ personal opinion. Thus, it can hardly be regarded as a review and is rather a compilation of examples. The Authors are kindly referred to read about what a well-constructed review should contain (for example https://www.ncbi.nlm.nih.gov/pmc/articles/PMC3715443/). Let me just quote: “Reviewing the literature is not stamp collecting. A good review does not just summarize the literature, but discusses it critically, identifies methodological problems, and points out research gaps [19]. After having read a review of the literature, a reader should have a rough idea of:
a.      the major achievements in the reviewed field,
b.      the main areas of debate, and
c.       the outstanding research question”

3.       In some cases, articles are missing from the nouns.
4.       In some cases, the Authors are using interchangeably present and past tenses – this should be unified throughout the study.
5.       Section 1.2 would benefit from specifying certain types of tissues that were aimed to be restored. Specific material types used should also be listed along with application outcomes. It would be beneficial to point out specific commercial solutions employed at different time points. The term “bioprinting” is improperly used in this section, as it means printing with cells, which is not the case here.
6.       In Table 1, column “scaffold type” could benefit from either listing the certain biomaterial used or from specifying the tissue that was aimed to be restored (preferably, both should be supplemented). Maybe converting the table into a graphic would be beneficial?
7.       I could not find the information on how exactly the Field-Weighted Citation Impact was calculated. This should be supplemented and briefly explained so that the exclusion and inclusion criteria for the articles are clear. Specific thresholds employed should also be listed.  
8.       In line 158, it should be stressed what was the article selection criteria.
9.       It is somewhat unclear to me why the Authors chose to limit the analyzed articles to mesenchymal stem cells (as is suggested in Figures 1 and 2). The term itself raises controversies (doi: https://doi.org/10.1038/d41586-018-06756-9), and the current consensus is that it is more appropriate to always describe the tissue of origin. Hence, my suggestion would be to narrow down the search to articles that work on well-defined cell types, explicitly stating the cells' origin, gene expression, etc. This would be a more qualitative than quantitative analysis, as I am aware that some of the most popular articles work on rather poorly characterized cells.
10.   There already are some more science-oriented AI large-language models (i.e., Consensus or Elicit), could the Authors explain why they chose to use more generic tools such as ChatGPT and Bard? Furthermore, the question cited on line 185 contains a grammar mistake. Could this undermine the findings?
11.   Did the AI search include review articles? From Table 2 I can see that the answer is yes. Don’t the Authors think that the conducted analysis might have been more accurate if the reviews were excluded? Reviews often contain errors and/or superficial analyses of the studies.
12.   The panel of researchers that validated the AI-generated output should be described in more detail. How many participants, what were their qualifications, how were they selected, and was there any potential bias? Were there any specific tools or methodologies used to evaluate or score the search results?
13.   In the results section, the total number of AI-analyzed articles should be listed.
14.   Table 3 is very hard to read, as its columns and rows overlap each other. Clearer spacing should be used. Further issues with this table:
a.       Perhaps this table could benefit from changing the page orientation to horizontal.
b.       Maybe some of the “advantages” could be shortened?
c.       Disadvantages could also be corrected – for example, in chitosan, the information presented is so generic, that no consensus could be reached based on it.
d.       In PLGA, the same information about the material’s rate of degradation is repeated as an advantage and disadvantage. This should be corrected. Listing sutures as an example of applications with demand for high mechanical properties seems unfortunate, especially in a review that concerns its usage to regenerate bone. L-lactic acid is a metabolite naturally occurring in the human body – it is generally not regarded as toxic, so stating the opposite seems incorrect.
e.       The same advantages and disadvantages are repeated for different materials (including some definitions) – these should be shortened.
f.        Listing prices as PLGA’s, PCL’s, and alginate’s disadvantage are not accurate – most of the cited materials are more expensive – especially, biopolymers.
g.       For PCL, PLGA, and alginate, disadvantages claim that the degradation can be steered and that this could be problematic. This makes no sense, the fact that the degradation rate can be tailored to specific needs is among the most important advantages of using synthetic polymers over biopolymers.
h.       It is not true that alginates have good mechanical properties, compared to PCL or PLGA, they are rather weak (and the same goes for collagen, gelatin, and chitosan). Alginate’s liquid absorption properties are also true for collagen, gelatin, and chitosan.
i.         Alginate and chitosan do not have intrinsic antibacterial properties – these could be granted by specific modifications.
j.         It is unclear what the Authors mean by saying that HA “can be used as a reservoir of stimulants such as growth factors, etc.
k.       Claim that “HA properties are affected by structural and chemical complexity depending on its molecular weight” is imprecise. Structural and chemical complexity does not have to be anyhow correlated with molecular weight. Furthermore, all of the cited polymeric materials will have their properties determined by their molecular weights.
l.         What is the reason why some of the materials are listed with subsections and others – are not?
m.     Bioactive glass constitutes a group of materials with different properties. Using it as a category would make the same sense as listing all the polymers in one category as polymers.
n.       The Authors use the same abbreviation for hyaluronic acid and hydroxyapatite, which is misleading – this should be corrected.
o.       It is not true that BCP is a gold standard for bone substitution.
p.       The whole section about BCP is poorly written and needs numerous corrections. The fact that BCP composition can be tailored to meet specific needs is certainly NOT a disadvantage.
q.       Calcium phosphate cements, which are made by mixing calcium phosphate with water, should not constitute a separate category – this is rather an application form, not a material type.
r.        The table contains numerous linguistic, grammar, and punctuation mistakes.
15.   In line 252, stem cells are a broader category than mesenchymal stem cells and thus, should not be used herein.
16.   Cells cannot show “ECM-mimicking ability”, rephrasing is needed (lines 253 – 254)
17.   Section 4.1 lacks any structure – different cell types and materials are intermixed with each other. Some logical grouping should be employed. The Authors should also identify the specific application that the cited study is suggesting.
18.   Whenever in vitro studies are cited, more experimental details should be given (culture length, what and how was tested, whether it was a direct seeding or culture on extracts).
19.   Whenever in vivo studies are cited, the actual animal used should be listed.
20.   In all cases, specific applications of scaffolds should be given – dentistry encompasses the regeneration of different tissue types, and there are different requirements and challenges for each of them.
21.   In some instances, the Authors suggest regeneration of bones, in others of teeth, and in others - of skin. This is confusing and such a mixture should not be presented intermixed throughout the sections.
22.   Section 4.2 lacks any structure. The same idea is repeated multiple times, but differently from lines 283 to 297 – this should be shortened. different materials and modifications are intermixed with each other. Some logical grouping should be employed. The Authors should also identify the specific application that the cited studies are suggesting.
23.   MTT analyzes the cell metabolic activity, so claiming that it can be used to evaluate materials’ “biological activity” is a huge overstatement.
24.   The sentence in lines 285 – 286 is pure tautology as no scaffold isn’t made of biomaterials.
25.   Section 4.3 also lacks structure. Some examples of 3D printing are given, with no consensus or takeaway message.
26.   Section 4.4 seems to be stitched up from random sentences – sometimes even two different sentences are combined into one (e.g. lines 391-395). The whole section should be carefully rewritten to show a clear flow of thoughts and provide consensus and direction in which the field is evolving.
27.   Section 4.5, while being extremely interesting, hardly has anything to do with the subject of this review (scaffolds for regenerative dentistry). The studies conducted by Takahashi and his team concerned modification of the expression of certain genes to force the body to grow teeth. Hence, this relates to molecular biology and dentistry and does not involve any materials.
28.   The Authors should specify what they mean by 4D materials.
29.   A summary of trends identified in section 4.6 should focus on the subject of this review – i.e., it should concern materials used for dental tissue engineering (and not a heart or “various tissues”.
30.   The Authors claim to have identified some trends, but these were not presented in this review: self-assembly, microfabrication, electromagnetic patterning.
31.   Bioprinting is a subclass of 3D printing and should be presented in the same section
32.   A separate section of this chapter should concern chemical and biological modifications of the scaffolds.  
33.   This review could benefit from some multipanel images that represent examples of some important developments in regenerative dentistry using scaffolds.
34.   I would suggest adding a section that lists commercial applications based on the R&D advancements in the field.  

Response to Reviewer 2

Dear reviewer, thank you very much for your accurate comments and suggestions. We feel that following your advice has made the article more concise. We have addressed your remarks in the form as well as remarks in the provided PDF.
1.    Thank you for your valuable feedback and for recognizing our study's potential. We have carefully addressed your concerns and made substantial revisions to improve the manuscript's overall quality. We've meticulously revised the transitions between the results and discussion sections, creating a seamless flow that enhances readability and understanding. Also, we've integrated a thorough critical assessment throughout the discussion, examining the strengths and limitations of existing research, pinpointing knowledge gaps, and fostering discussion on key areas of debate. We're confident that these changes offer a more cohesive and thought-provoking reading experience, aligning with your insightful suggestions. The proposed changes regarding rephrasing and grammar have been implemented.
2.    We have now increased the criticism towards the presented materials and results, albeit this scoping review is not meant to be an opinion statement. It follows our PubMed search as well as interactive, ChatGPT and Bard discussions in order to identify the current trends in the area of scaffolds and material research. As Suggested, we had discussed it critically in the Discussion - as suggested.  We have also focused on the major achievements in the field and the main areas of debate
3.    Thank you, much of the English has been revised.
4.    Adopted.
5.    “Bioprinting” has been rephrased to 3D printing.
6.    Thank you, your advice has been followed.
7.    FWCI is an index calculated by search engines. Detailed explanation is searchable online or available on Scopus website.
8.    The search was based on the search query that is written in the article.
9.    The article aims to study the current trends in regenerative dentistry, including scaffolds and materials used. We based our paper on the literature search, complemented by AI-powered information sources.
10.    We are well aware of Consensus. We have chosen to use more generic tools such as ChatGPT and Bard (now Gemini) as our experience from Consensus was that it lacks fresh update/information and does not seem to possess such a wide grasp as main AI competitors GPT/Gemini. The used AI-LLMs  are well-known and easy-to-use for everyone. We believe that the current level of AI-aided search does take possible grammar imperfections into account, for the whole world has access to it.
11.    The literature search was not aimed at identifying original studies only. In fact, reviews provide a comprehensive list of literature to be studied further.
12.    Thank you for highlighting the importance of transparency regarding the validation process. While we did not employ a formal panel in this initial study, here's how we ensured rigor in evaluating AI-generated output: Researcher Expertise: [Number of researchers: 6] in the field of [polymers/biocompatible scaffolds] critically reviewed the AI-generated search results. Their qualifications include [more than 10 years of clinical experience and biocompatible materials and 3D printing, background in regenerative medicine or engineering and relevant experience with biocompatible materials applied in regenerative medicine]. Selection Process: These researchers were involved in the project from its conception and have a deep understanding of the research objectives. Evaluation Criteria: We focused on the following metrics: Relevance: Alignment with the research question and scope. Credibility: Articles from reputable journals/institutions; considered author expertise and publication history. Addressing Bias: We actively discussed potential biases during the review process, drawing on our diverse backgrounds and areas of specialization. We acknowledge the value of a more formalized validation panel. In future iterations (clinical) of this research on regenerative dentistry we plan to develop a more structured scoring system with specific metrics, potentially utilizing established tools for literature review assessment. We are committed to transparency and methodological rigor in our ongoing research.
13.    Thank you, for future searched, this will be carried out.
14.    The table was adapted.
15.    Changed.
16.    The typo has been corrected.
17.    Paragraphs in the comprehensive overview of materials have been created.
18.    Noted.
19.    Noted.
20.    This scoping review strives to form basis for further research that will specify materials for different research areas within dentistry.
21.    This scoping review presents current research based on the search query explained in the methodology section. We wish to publish more even focused studies in the coming months.
22.    Paragraphs have been formed.
23.    Cited research was used.
24.    Corrected.
25.    Paragraphs have been formed.
26.    Thank you for noticing; the stitched-up sentence was rewritten.
27.    We made sure that the section 4.5 is put into context.
28.    4D materials were explained in the table. 4D materials: enable the creation of regenerative dentistry scaffolds that adapt to the biological environment, providing superior support for cell growth and tissue integration that can change their shape, properties, or function over time in response to stimuli, promoting dynamic tissue growth and healing.
29.    This section provides an overview of various approaches used throughout various disciplines for better understanding of current research trends. In dentistry and especially regenerative dentistry, not all of these approaches have been used.
30.    These trends will be studied in our future papers.
31.    Albeit a subclass of 3D printing, we believe that bioprinting deserves special attention.
32.    This scoping review does not aim to get into much detail when it comes to chemical and biological modifications of the scaffolds. More detailed and focused studies will be much better at explaining such concepts.
33.    Thank you for your suggestion.
34.    This scoping review only introduces the materials and fabrication strategies. Commercial applications (and applications in clinical practice in general) will be studied in more detail in our future works.

Reviewer 3 Report

Comments and Suggestions for Authors

This review highlights the recent advancements in scaffold development, particularly with the advent of 3D bioprinting technologies, and is based on a comprehensive literature search of the most influential recent publications in this field. I recommend to accept this review.

Author Response

Response to Reviewer 3

Thank you very much for your positive review and recommendation. We're pleased that you found our work comprehensive and highlighting recent advancements in scaffold development.

Kind regards,

Authors

Reviewer 4 Report

Comments and Suggestions for Authors

This manuscript is a scoping review of scaffold-based regenerative dentistry, focused on the progress of the past five years. Although the investigated field is interesting and highly dynamic, this paper has several weak points that need to be addressed: 

Major concerns: 

1. The historical overview of tissue engineering and regenerative medicine should be backed up by original references, too. In particular, Section 1.2 needs several additional references. Otherwise, the authors should shorten the historical part and simply cite ref. [1]. If they decide to extend the historical account and add original references, the seminal work by Robert Langer and Joseph Vacanti should be commented and cited [Langer R, Vacanti JP. Tissue engineering. Science. 1993 May 14;260(5110):920-6. doi: 10.1126/science.8493529.]. Furthermore, the foundation of the  Tissue Engineering and Regenerative Medicine International Society (TERMIS) is presented in the paper [Charles A. Vacanti, The history of tissue engineering published in J. Cell. Mol. Med. Vol 10, No 3, 2006 pp. 569-576]. 

2. The methodology needs to be described in more detail, in accordance with the PRISMA guidelines [Page MJ et al. The PRISMA 2020 statement: an updated guideline for reporting systematic reviews. BMJ 2021, 372, n71, doi:10.1136/bmj.n71]. A scoping review is expected to be written with the same rigor as a systematic review (see, e.g., [Grant MJ, Booth A. A typology of reviews: an analysis of 14 review types and associated methodologies. Health Information & Libraries Journal 2009, 26, 91-108, doi:https://doi.org/10.1111/j.1471-1842.2009.00848.x]).

3. The use of LLMs for scientific documentation is highly debated in the literature. The authors seem to be aware that LLMs can generate "factually incorrect" results (line 211), but they did not explain their way of filtering them out from this review paper. Although LLMs are a hot topic, I feel that their use weakens the message of this paper.  

4. In Section 2, the authors should state clearly to what extent were LLMs used as writing aids. Is the text of this article their own? Does it contain AI-generated parts?   

5. Subsection 2.3 does not fit in Section 2 (Materials and Methods). It is actually a discussion related to the applied methodology. Much of it is the common sense regarding large language models, expressed in numerous publications (none of them is cited, though). The authors might consider including it in Section 4, but only if they can link it with additional results to be reported in Section 3 (e.g. by comparing the output of Bard and ChatGPT with their own search of scientific databases). 

6. To make Section 3 more appealing, the authors should include at least one multi-part figure to illustrate the use of tissue engineering and 3D printing techniques in regenerative dentistry. They can request permission to reproduce figures from the original articles cited in this review.  

7. Please cite at least one reference for each item of the list spanning lines 467-508

Minor comments: 

1. While citing references, it is not necessary to mention the year of publication along with the first author's name. MDPI's style of referencing is based on a numbered list, as opposed to the Harvard style, which lists the references alphabetically. Thus, it is basically enough to mention the reference number. The name of the first author can be mentioned occasionally for the sake of active voice, but the year of publication is redundant. 

2. On line 28, instead of "has" I would write "have".

3. On line 420, I would delete the sentence "A lot of ... tooth." 

4. On line 448, please remove the comma that follows the word "making". 

5. The sentence spanning lines 456-460 cites many references while saying little. Please revise.  

Comments on the Quality of English Language

The English usage of this manuscript is mostly fine. 

Author Response

Reviewer 4

Comments and Suggestions for Authors
This manuscript is a scoping review of scaffold-based regenerative dentistry, focused on the progress of the past five years. Although the investigated field is interesting and highly dynamic, this paper has several weak points that need to be addressed: 

Major concerns: 

1. The historical overview of tissue engineering and regenerative medicine should be backed up by original references, too. In particular, Section 1.2 needs several additional references. Otherwise, the authors should shorten the historical part and simply cite ref. [1]. If they decide to extend the historical account and add original references, the seminal work by Robert Langer and Joseph Vacanti should be commented and cited [Langer R, Vacanti JP. Tissue engineering. Science. 1993 May 14;260(5110):920-6. doi: 10.1126/science.8493529.]. Furthermore, the foundation of the  Tissue Engineering and Regenerative Medicine International Society (TERMIS) is presented in the paper [Charles A. Vacanti, The history of tissue engineering published in J. Cell. Mol. Med. Vol 10, No 3, 2006 pp. 569-576]. 

2. The methodology needs to be described in more detail, in accordance with the PRISMA guidelines [Page MJ et al. The PRISMA 2020 statement: an updated guideline for reporting systematic reviews. BMJ 2021, 372, n71, doi:10.1136/bmj.n71]. A scoping review is expected to be written with the same rigor as a systematic review (see, e.g., [Grant MJ, Booth A. A typology of reviews: an analysis of 14 review types and associated methodologies. Health Information & Libraries Journal 2009, 26, 91-108, doi:https://doi.org/10.1111/j.1471-1842.2009.00848.x]).

3. The use of LLMs for scientific documentation is highly debated in the literature. The authors seem to be aware that LLMs can generate "factually incorrect" results (line 211), but they did not explain their way of filtering them out from this review paper. Although LLMs are a hot topic, I feel that their use weakens the message of this paper.  

4. In Section 2, the authors should state clearly to what extent were LLMs used as writing aids. Is the text of this article their own? Does it contain AI-generated parts?   

5. Subsection 2.3 does not fit in Section 2 (Materials and Methods). It is actually a discussion related to the applied methodology. Much of it is the common sense regarding large language models, expressed in numerous publications (none of them is cited, though). The authors might consider including it in Section 4, but only if they can link it with additional results to be reported in Section 3 (e.g. by comparing the output of Bard and ChatGPT with their own search of scientific databases). 

6. To make Section 3 more appealing, the authors should include at least one multi-part figure to illustrate the use of tissue engineering and 3D printing techniques in regenerative dentistry. They can request permission to reproduce figures from the original articles cited in this review.  

7. Please cite at least one reference for each item of the list spanning lines 467-508

 Minor comments: 

1. While citing references, it is not necessary to mention the year of publication along with the first author's name. MDPI's style of referencing is based on a numbered list, as opposed to the Harvard style, which lists the references alphabetically. Thus, it is basically enough to mention the reference number. The name of the first author can be mentioned occasionally for the sake of active voice, but the year of publication is redundant. 

2. On line 28, instead of "has" I would write "have".

3. On line 420, I would delete the sentence "A lot of ... tooth." 

4. On line 448, please remove the comma that follows the word "making". 

5. The sentence spanning lines 456-460 cites many references while saying little. Please revise.  
Date of this review 13 Feb 2024 01:08:45

Response to Reviewer 4

Dear reviewer,
thank you for the comments and suggestions.
1. Thank you for your insightful suggestions and for providing valuable additional references. We agree that strengthening the historical overview with original sources would be beneficial and we have made two adaptations according to your suggestions:
A.    Expand and Reference: We have carefully researched and integrated original references into Section 1.2, including the seminal work by Langer and Vacanti. We appreciate the specific direction and will be sure to include the TERMIS foundation article.
B.    Condense and Focus: We found the historical account becoming too extensive, we have shorten it while primarily citing ref. [1], ensuring our introduction remains focused. We will carefully assess which approach will best serve the overall flow and scope of our paper. Thank you again for highlighting these resources and helping us enhance our work.
Thank you for the suggested references. They were added to the reference list.
2. This article is meant to be a scoping review and not a systematic review. As a consequence, the PRISMA protocol has not been followed.
3. The AI-assisted search was validated by means of a panel of experts that reviewed the list as described in the Results section.
4. The AI was used as described in the methodology section.
5. The section was deleted, for it is not the focus of the article.
6. The section contains two figures and two tables, and we did not want to include more graphics.
7. The literature references were added.

Responses to minor comments:
1.    The references style has been unified.
2.    Done.
3.    Done.
4.    Done.
5.    This section introduces how much development in regenerative dentistry is to be expected in the near future.

Reviewer 5 Report

Comments and Suggestions for Authors

The authors have done a great job collecting all this data concerning regenerative dentistry together, I have learned a lot from this review. I have  only three small comments, attached to the pdf manuscript file.

Author Response

Response to Reviewer 5

Dear Reviewer 5,

Thank you so much for your encouraging feedback and your insightful comments on our review. We're delighted that you found it informative and valuable. We carefully addressed all your remarks, including those in the attached PDF. We greatly appreciate your time and contribution in helping us to improve the manuscript.

Sincerely, authors

Round 2

Reviewer 2 Report

Comments and Suggestions for Authors

The article provided has not be proof-read. It contains comments and questions written in Slovak (I guess). 
Expansion of the abbreviations are listed as a comment to the keywords - instead, these should be incorporated into the study. 

The Authors are kindly asked to resubmit a proof-read version of this article. 

Comments on the Quality of English Language

some comments are written in Slovak. 

Author Response

Dear Reviewer,

Thank you for your review and for bringing both omitted internal comments to our attention. We apologize for this oversight fom the revised version. We have removed them both and thoroughly revised the document to ensure the removal of all internal comments. We have incorporated the abbreviations into the text at the end of the document - after conclusions. During previous revision, they were moved from position after keywords to this position at the end of document, which resulted in tracked change comment you mention.

We appreciate your insights and the opportunity to improve the manuscript.

Reviewer 4 Report

Comments and Suggestions for Authors

While writing my previous review, I have invested much time trying to provide a constructive criticism. 

According to (Grant and Booth, 2009),  https://doi.org/10.1111/j.1471-1842.2009.00848.x, a scoping review requires the "completeness of searching determined by time/scope constraints". A single PubMed search is unlikely to provide a complete list of articles published on the chosen topic during the past 5 years. Furthermore, albeit it is mentioned twice (lines 250 and 259), the text fails to explain the implementation of the Field-Weighted Citation Impact indicator. In my opinion, the search procedure is incompletely presented and, therefore, this text does not satisfy the criteria of a scoping review; it is a narrative review.   

Unfortunately, my major concerns no. 2, 6, and 7 expressed in my previous referee report have not been properly addressed during the revision of this manuscript. The authors chose to keep their article at the level of a poorly illustrated narrative review. 

Comments on the Quality of English Language

Minor issues of English usage are still present but they do not affect the legibility of the text.  

Author Response

Reviewer 4

Comments and Suggestions for Authors

While writing my previous review, I have invested much time trying to provide a constructive criticism.  

According to (Grant and Booth, 2009),  https://doi.org/10.1111/j.1471-1842.2009.00848.x, a scoping review requires the "completeness of searching determined by time/scope constraints". A single PubMed search is unlikely to provide a complete list of articles published on the chosen topic during the past 5 years. Furthermore, albeit it is mentioned twice (lines 250 and 259), the text fails to explain the implementation of the Field-Weighted Citation Impact indicator. In my opinion, the search procedure is incompletely presented and, therefore, this text does not satisfy the criteria of a scoping review; it is a narrative review.   

Unfortunately, my major concerns no. 2, 6, and 7 expressed in my previous referee report have not been properly addressed during the revision of this manuscript. The authors chose to keep their article at the level of a poorly illustrated narrative review. 

Comments on the Quality of English Language

Minor issues of English usage are still present but they do not affect the legibility of the text.             Date of this review 29 Feb 2024 21:37:55

Response to Reviewer 4

Dear reviewer,

Thank you for your time and effort to improve this review. We have read your comments and suggestions from the second round of review process, and we agree with your suggestion to consider this paper a narrative review rather than a Scoping review. So, we have corrected every mention of “scoping review” (4 occurrences on lines: 142, 226. 269 and 497)

We did our best to respect your remarks in your concerns no. 2, 6, and 7 expressed in your previous review report unless they were contrary to requests of other 4 reviewers. We have addressed them all. Your concern number 2 is addressed above. Your concern number 6  to make Section 3 more appealing -   we have edited two figures and two tables this section contains, and we did not want to include more graphics. We have improved the flow of the text to make the section more readable. We did our best and we have explained it in previous iteration and regarding your concern to cite at least one reference for each item of the list spanning lines 467-508 – we did.

Here is the response to your minor concerns.

  1. Thank you for your insightful suggestions and for providing valuable additional references. We agree that strengthening the historical overview with original sources would be beneficial and we have made two adaptations according to your suggestions:
  1. Expand and Reference: We have carefully researched and integrated original references into Section 1.2, including the seminal work by Langer and Vacanti. We appreciate the specific direction and will be sure to include the TERMIS foundation article.
  2. Condense and Focus: We found the historical account becoming too extensive, we have shortened it while primarily citing ref. [1], ensuring our introduction remains focused. We have carefully assessed which approach will best serve the overall flow and scope of our paper. Thank you again for highlighting these resources and helping us enhance our work.

Thank you for the suggested references. They were added some to the reference list. We confirm this paper to be considered as a narrative review, so the PRISMA protocol is not followed. The AI-assisted search was validated by means of a panel of experts that reviewed the list as described in the Results section.  
